# Effect of Chemical Fertilizer with Compound Microbial Fertilizer on Soil Physical Properties and Soybean Yield

Chenye Fu [1,2], Weiran Ma [1], Binbin Qiang [1], Xijun Jin [1], Yuxian Zhang [1,3] and Mengxue Wang [1,*]

1. School of Agronomy, Heilongjiang Bayi Agricultural University, Daqing 163000, China; 18080915925@163.com (C.F.); 13100855125@139.com (W.M.); qiangbb2022@163.com (B.Q.); shaoxiang1979@163.com (X.J.); zyx_lxy@126.com (Y.Z.)
2. College of Environment, Sichuan Agricultural University, Chengdu 611130, China
3. National Multigrain Engineering and Technology Center, Daqing 163000, China
* Correspondence: wangmengxue1978@163.com; Tel.: +86-138-3684-2844

**Abstract:** Compound microbial fertilizer is a new type of environmentally friendly slow-release fertilizer that can effectively improve the physical and chemical properties of the soil, significantly improve the ecological environment, and promote the sustainable development of agriculture. In this study, we conducted a field experiment to evaluate the impact of different applications of chemical fertilizer combined with composite microbial fertilizer on soil physical properties and soybean yields at Heshan Farm, Heilongjiang Province, China, during 2021–2022. Soybean varieties "Jinyuan 55" and "Keshan 1" were treated with three treatments implemented as follows: T1 (conventional fertilization), T2 (50% N fertilizer + compound microbial fertilizer), and T3 (0 N fertilizer + compound microbial fertilizer). Compared to conventional fertilization (T1 treatment), the application of composite microbial fertilizers (T2 and T3) resulted in a decrease in soil bulk density and an increase in porosity. Notably, we observed that moderate application of the composite microbial fertilizer (T3) led to a decrease in the volume fraction of clay particles and an increase in the volume fraction of sand particles. Furthermore, all treatments exhibited high content of agglomerates larger than 5 mm at 0–20 cm. The application of composite microbial fertilizers (T2 and T3) promoted the formation of large soil agglomerates and reduced the presence of micro-agglomerates smaller than 0.25 mm. In 2021–2022, The soybean yield increased by 13.02% in the T2 treatment compared with the T1 treatment and decreased by 9.34% in the T3 treatment. We concluded that the appropriate application of compound microbial fertilizer can help protect black soil, enhance the self-repair capability of black soil, and improve soybean quality in abnormal precipitation years. These results provide an actionable basis for constructing and developing green fertilizer systems for the soybean industry.

**Keywords:** complex microbial fertilizer; soybean; soil physical properties; yield

## 1. Introduction

In China, with soybean [1], there is a historical record of its cultivation [2] dating from as early as 4700 years ago. Soybean is one of the world's important oil and grain crops, widely grown in the northeast region of China, Inner Mongolia, Shandong, Henan, Hebei, and other regions [3]. An indispensable part of the soybean production process is the application of fertilizer [4], which can play a direct role in the improvement of soybean yield and quality. However, the uncontrolled and meaningless application of fertilizer produces a series of problems, such as a decline in the fertilizer utilization rate, cracking and crumbling of the soil, contamination of rivers and groundwater, and a decline in soil fertility [5], which may even lead to unsatisfactory soybean quality and production efficiency. Currently, overexploitation of land has led to soil degradation to varying degrees [6]. Soil degradation leads to environmental problems such as reduced soil fertility, nutrient depletion, and soil salinization, limiting the productivity of agricultural land [7]. Fertilizer application is an

important means of improving crop yield and quality, however, conventional fertilizers are characterized by high solubility, low thermal stability, and low molecular weight, while most of the nutrients are lost through surface runoff, leaching, and volatilization, leading to problems such as eutrophication of water bodies. At the same time, the high salt content of traditional fertilizers tends to lead to soil compaction and reduce soil quality when used for a long period of time [8,9]. In view of this severe environmental impact, China's 14th Five-Year Plan proposes to carry out in-depth soil testing and formulation for fertilizer application, continuously optimize the structure of fertilizer inputs, increase the use of organic fertilizers, and promote efficient fertilizer application techniques. Therefore, there is an urgent need to support and develop sustainable agriculture through a new type of environmentally friendly, slow-release fertilizer [10,11]. The use of a compound microbial fertilizer can effectively improve the soil. Tu Yongcheng [12] found that compound microbial fertilizer can increase the water content of the soil, reduce the soil's bulk weight, and increase the soil's porosity. Li Chang et al. [13] studied the effect of microbial fertilizers on soil physical properties and found that microbial fertilizers can significantly improve the physical properties and ecological environment of the soil, while the relative water content and soil temperature of the soil increased significantly, and the soil capacity of the working layer from 0 to 20 cm decreased significantly. Generally speaking, the use rate of fertilizer is inversely proportional to the application rate. After microbial fertilizer is applied to the soil, beneficial microbial strains and some colloidal-like substances in the soil combine with each other and plant mucus [14] to promote soil agglomeration and improve the soil aggregation structure, which alleviates to a certain extent the problems of soil slumping, drought, poor aeration, and loss of soil particles [15]. The studies of Rasool R et al. [16] and Wilson WS [17] showed that the organic materials significantly improved the total porosity of the soil. Wang Fengjun et al. [18] showed that the use of fertilizers containing organic matter in celery fields increased total soil porosity and soil water-stable aggregate content. However, different composite microbial fertilizers or different application times may have led to variations in the results of the experiment. Different zoning factors in various regions led to mixed results in the experiment.

This study focuses on soybean, a major economic food crop in Heilongjiang Province, China. The aim was to investigate the effects of compound microbial fertilizer on soybean growth and soil physical properties. Field trials were conducted using different application gradients of compound microbial fertilizer in combination with chemical fertilizer. The goal was to promote the use of chemical fertilizer with compound microbial fertilizer in soybean cultivation, establish a green fertilizer system, enhance resource use efficiency, improve the competitiveness of the soybean market in China, and contribute to a cleaner production environment and a stronger ecological self-healing capacity. The findings of this study will also support the greening and low-carbonization of China's rural production methods.

## 2. Materials and Methods

### 2.1. Introduction to the Experimental Site

This experiment was conducted in the experimental field of Heshan Farm, Jiusan District, Heilongjiang Province, China. The location of the field is at latitude 48°43′–49°03′ N and longitude 124°56′–126°21′ E (Figure 1). The area has a cold-temperate continental climate with an average annual temperature of 1.8 °C, an annual effective cumulative temperature of 2000–2450 °C, an average annual rainfall of 550 mm, and an average annual frost-free period of 121 days (Figure 2). The soil type in the experimental area is black soil, and the basic physical and chemical properties of the 0–20 cm soil layer are presented in Table 1. The soil in this area is weakly acidic.

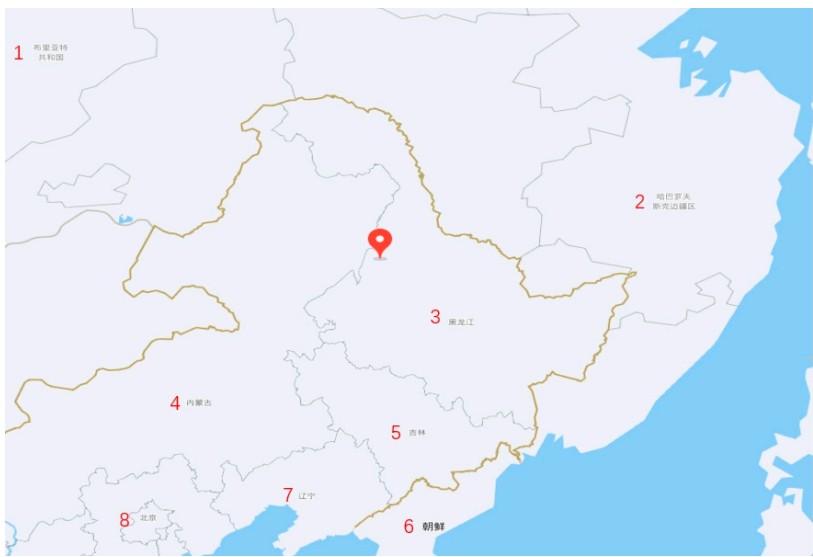

**Figure 1.** Map showing the geographical location of the test site. Note: The red numbers on the map indicate the location of the districts, 1 for the Republic of Buryatia, 2 for the Khabarovsk Krai of Russia, 3 for Heilongjiang Province of China, 4 for the Inner Mongolia Autonomous Region of China, 5 for Jilin Province of China, 6 for the State of North Korea, 7 for the Liaoning Province of China, and 8 for the Beijing Municipality of China.

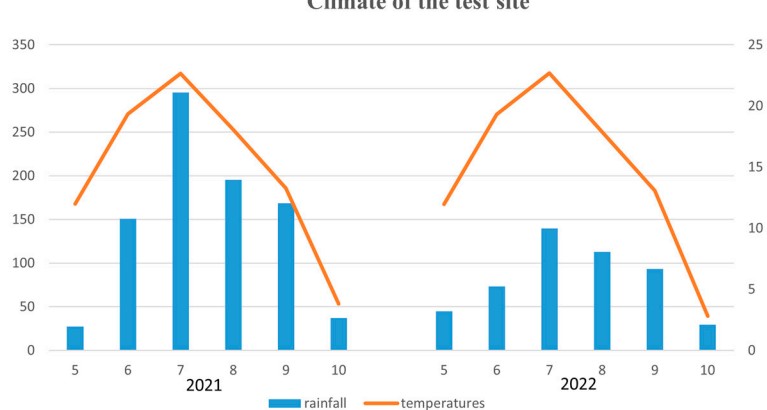

**Figure 2.** Temperature and rainfall (2021–2022).

**Table 1.** Basic physical and chemical properties of the soil 0–20 cm before the test.

| Years | pH | Alkali-Hydrolyzed Nitrogen (mg/kg$^{-1}$) | Available Phosphorus (mg/kg$^{-1}$) | Quickly Available Potassium (mg/kg$^{-1}$) | Organic Matter (g/kg$^{-1}$) | Bulk Density (g/cm$^{-3}$) |
|---|---|---|---|---|---|---|
| 2021 | 6.30 | 141.80 | 30.60 | 180.00 | 17.94 | 1.26 |
| 2022 | 6.14 | 137.92 | 21.79 | 177.35 | 15.30 | 1.18 |

*2.2. Test Materials*

The high-protein variety Jinyuan 55 (P1) and the high-fat variety Keshan 1 (P2) were selected as the test varieties.

Jinyuan 55 (P1) has an average fertility period of 115 days for spring sowing in the northern region. It is characterized by white flowers, an astringent plant shape, and a limited pod-rich habit.

Keshan 1 (P2) has a fertility period of 112 days, with long leaves, purple flowers, and a sub-limited pod habit.

The locally applied conventional fertilizer rates were as follows: 120 kg/hm$^{-2}$ of urea (46% N), 150 kg/hm$^{-2}$ of calcium superphosphate (46% $P_2O_5$), and 30 kg/hm$^{-2}$ of potassium sulfate (50% $K_2O$).

The compound microbial fertilizer used in this study was provided by Heilongjiang Zhaofeng Hemei Company. It had an organic matter content of at least 20% and a total nutrient content (N + $P_2O_5$ + $K_2O$) of 8%, with a ratio of 2:2:1. The specific fertilizer used was F01 (all-purpose) compound microbial fertilizer, which contained various bacteria species such as Bacillus subtilis, Bacillus licheniformis, Bacillus spp., Bacillus megaterium, Bacillus coli, Bacillus sphaericus, special functional bacteria, and carrier. The effective number of live bacteria in the fertilizer was at least 0.2 billion per gram.

In this trial, the previous crop was maize to avoid the hazards of contiguous cropping, and no other experimental design was conducted on the previous crop to ensure the stability of the soil in each plot.

### 2.3. Experimental Design

The trial ran from 28 April 2021, until 18 October 2022. The trial used a one-way totally randomized block design with three two-variable treatments. Each treatment was replicated four times on a total of 24 plots. The plot comparison method was used, with each plot set up as an 8-row area. The row length was 6 m, the row width was 0.65 m, and the plot area was 31.2 m$^{-2}$. The soybean planting density was 35–400,000 plants-hm$^{-2}$. Based on the local fertilizer level, each hectare received 54 kg of N, 67.5 kg of $P_2O_5$, and 37.5 kg of $K_2O$. The treatments were designated as follows: Treatment T1 (conventional fertilizer application without the addition of composite microbial fertilizer), Treatment T2 (50% N fertilizer + compound microbial fertilizer), and Treatment T3 (0 N fertilizer + compound microbial fertilizer). The fertilizer was applied manually as a one-time basal application. Please refer to Table 2 for the specific experimental treatments. The crop was grown rainfed with no irrigation for all of the experimental years. In addition, pesticide and insecticide management were carried out according to local practices.

**Table 2.** Specific fertilization measures.

| Process Name | Urea kg/hm$^{-2}$ | Heavy Superphosphate kg/hm$^{-2}$ | Potassium Chloride kg/hm$^{-2}$ | Compound Microbial Fertilizer kg/hm$^{-2}$ | Total Nitrogen kg/hm$^{-2}$ | Total Phosphorus Content kg/hm$^{-2}$ | Total Potassium kg/hm$^{-2}$ |
|---|---|---|---|---|---|---|---|
| T1 (CK) | 120 (54) | 146.74 (67.5) | 75 (37.5) | 0 | 54 | 67.5 | 37.5 |
| T2 | 60 (27) | 88.04 (40.5) | 48 (24) | 2023.95 (1.3:1.3:0.6) | 54 | 67.5 | 37.5 |
| T3 | 0 (0) | 9.78 (4.5) | 18 (9) | 4048.05 (1.3:1.3:0.6) | 54 | 58.5 (19.57) | 36.0 (3) |

Note: The table shows the contents of N, $P_2O_5$, and $K_2O$ in the fertilizer. After the reduction of the fertilizer, the composite microbial fertilizer was used to make up for the lack of phosphorus and potassium in the T3 treatment. The amount of fertilizer below the parentheses is adjusted to ensure that the contents of N, $P_2O_5$, and $K_2O$ in the three treatments remain unchanged despite the change in fertilizer application. The ratio of nitrogen, phosphorus, and potassium elements is indicated in brackets under the composite microbial fertilizer.

### 2.4. Measurement Items and Methods

2.4.1. Determination of Soil Physical Properties

(1)    The determination of soil water content was conducted at different stages of soybean growth. Soil samples were collected by taking 10–20 cm of soil with a shovel during the full flowering and podding stage, mid-bulging stage, and maturity stage. Random-sized soil samples were weighed and recorded on-site, then placed in plastic bags and sealed for transportation. The soil samples were dried and weighed in an oven at 105 ± 1 °C to calculate the soil water content. Five replicates of samples were taken for each treatment.

(2) The soil capacity and porosity were determined using the ring knife method at a depth of 0–10 cm for both soybean seedling and maturity stages. Five replicates were taken for each treatment.

(3) To determine the soil mechanical composition, the PARIOPlus soil particle size automatic analyzer (Beijing Litai Science and Technology Co., Ltd., Beijing, China) was utilized. It determined the content of soil viscous particles and powder particles. Additionally, the content of soil sand particles was determined using the wet sieve method [19,20]. The soil particle size grading was carried out according to international system standards.

(4) Water stability aggregates were determined using the wet sieve method [21]. Soil samples obtained by dry sieving were mixed proportionally into 100 g soil samples. These soil samples were then placed on a set of sieves with sizes of 5 mm, 2 mm, 1 mm, 0.5 mm, and 0.25 mm. The sieving process was conducted using the DS/TTF-100 Soil Aggregate Analyser (Beijing Ding Sheng Rong He Science and Technology Co., Ltd., Beijing, China). The aggregate content of each particle size was determined after sieving. The agglomerates of each particle size were washed into a beaker using a spray bottle and then placed in an oven at 40 °C until a constant weight was achieved. The weight of the agglomerates was then measured. The percentage content of water-stable agglomerates for each particle size was calculated, with three replications for each treatment.

### 2.4.2. Measurement of Yield and Yield Components

At the maturity stage of soybean, one soybean plant per square meter was selected from each plot. Ten soybean samples with consistent growth were chosen to determine various yield components, including plant height, stem thickness, number of grains per plant, number of pods per plant, weight of grains per plant, and weight of 100 grains. These measurements were then used to calculate the actual yield of the plot, taking into account a water content of 13%.

### 2.5. Data Processing

Data processing was conducted using Excel 2016 software, while text editing and drawing of three-line tables were performed using Word 2016. One-way analysis of variance (ANOVA) was carried out using SPSS 22.0 software, and Duncan's test was used for post hoc analysis. GraphPad Prism 8 was utilized for plotting the data.

## 3. Results and Analyses

### 3.1. Effect of Chemical Fertilizer with Microbial Fertilizer on Soil Physical Properties

### 3.1.1. Effect of Chemical Fertilizer with Microbial Composite Fertilizer on Soil Water Content

The soil water content of different fertilizer treatments in 2022 is presented in Figure 3. During the soybean bloom period, the complex microbial fertilizer did not contribute to the soil water content. Compared with the Jinyuan 55 conventional fertilization (P1T1), the P1T2 treatment showed a significant decrease of 4.9% ($p < 0.05$) in soil water content compared to the T1 treatment. There was no significant difference between the composite microbial fertilizer treatments (T2, T3) and the conventional fertilization of Keshan 1 (P2T1) when compared to each other. However, at the full podding stage of soybean, the compound microbial fertilizer began to contribute to the increase in soil water content. The P1T2 treatment exhibited a significant rise of 18.34% in soil water content compared to the T1 treatment ($p < 0.05$), surpassing the Jinyuan 55 conventional fertilizer application (P1T1).

Compared to the conventional fertilizer application of Keshan 1 (P2T1), neither of the composite microbial fertilizer treatments (T2 and T3) showed significant differences from each other. During the middle of soybean bulging, the compound microbial fertilizer continued to promote the increase in soil water content. At the pod stage, compared to Jinyuan 55 conventional fertilization (P1T1), the P1T2 treatment significantly increased soil

water content, with a rise of 8.92% compared to the T1 treatment ($p < 0.05$). There were no significant differences observed among the Keshan 1 treatments. However, during the soybean maturity period, the promotion effect of compound microbial fertilizer on soil water content tended to plateau, and there were no significant differences observed among the different varieties and treatments. This could be attributed to the sampling time and climate conditions.

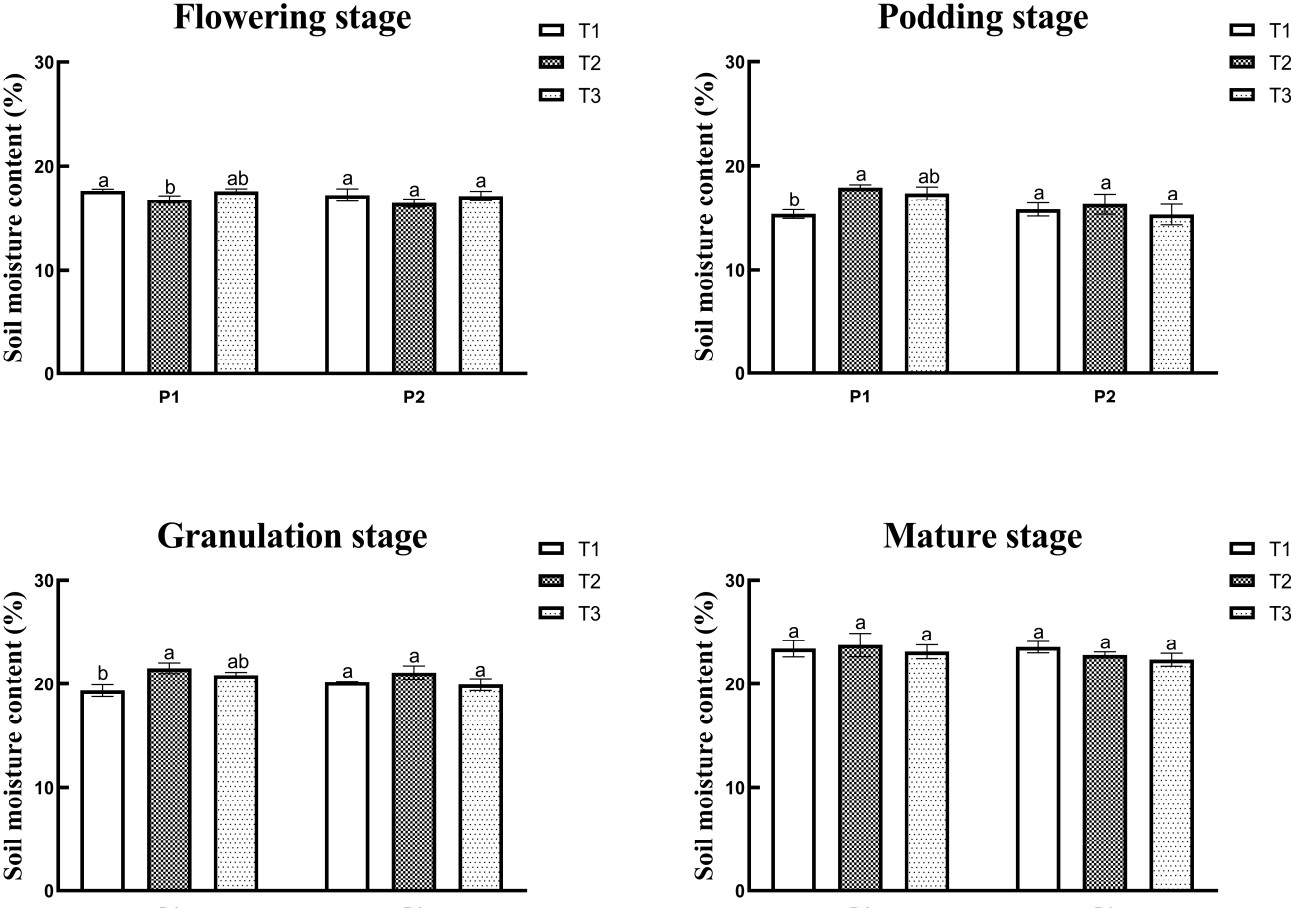

**Figure 3.** Effect of chemical fertilizers with microbial fertilizer complexes on soil moisture. Note: The icons P1 and P2 represent Jinyuan 55 and Keshan 1, respectively. The icons T1, T2, and T3 represent conventional fertilizer, 50% N fertilizer + 50% compound microbial fertilizer, and 0 N fertilizer + 100% compound microbial fertilizer, respectively. The lowercase letters indicate a significant difference between the treatments ($p < 0.05$).

3.1.2. Effect of Compound Microbial Fertilizers with Chemical Fertilizers on Soil Bulk Density

The impact of the 2022 chemical fertilizer combined with compound microbial fertilizer on soil bulk weight is illustrated in Figure 4. During the soybean seedling stage, it was observed that the soil bulk weight was significantly higher in the treatment without nitrogen fertilizer (P1T3) compared to the conventional fertilizer application (P1T1) in Jinyuan 55. The bulk weight increased by 17.97% compared to the T1 treatment ($p < 0.05$).

At soybean maturity, the complex microbial fertilizer was found to decrease soil bulk weight. The 1/2 nitrogen fertilizer treatment (P1T2) showed a significantly smaller soil capacity weight compared to the T1 treatment, with a reduction of 15.25% ($p < 0.05$). For the Keshan 1 variety, there was no significant difference in soil bulk weight between treatments at the seedling and maturity stages. The combined effect of compound microbial fertilizer and chemical fertilizer had a more pronounced impact on soil bulk density compared

to chemical fertilizer alone, indicating that the use of both can effectively reduce soil bulk density.

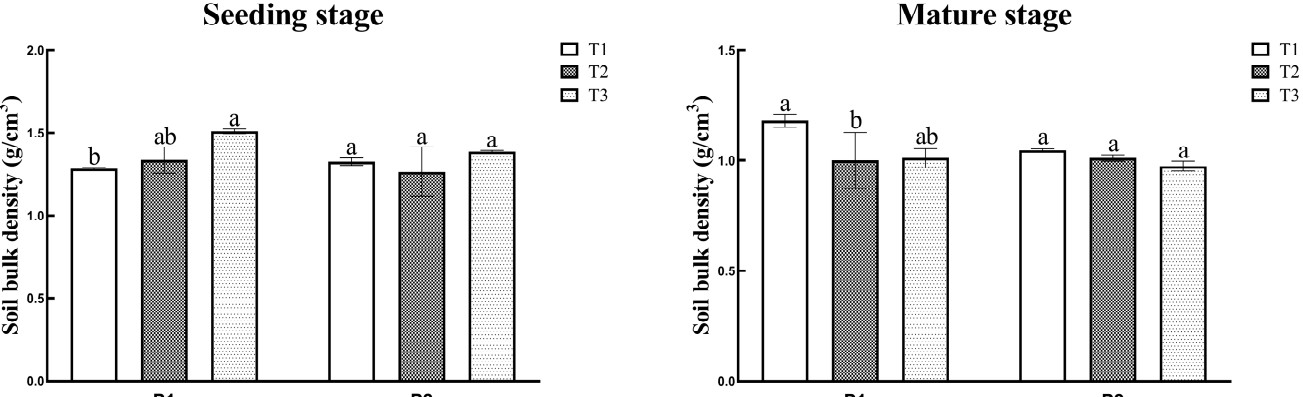

**Figure 4.** Effect of chemical fertilizers with microbial fertilizer complexes on soil capacity. Note: Different lowercase letters after a number indicate a significant difference (*p* < 0.05).

### 3.1.3. Effect of Chemical Fertilizers with Compound Microbial Fertilizers on Soil Porosity

In 2022, the soil porosity under the Keshan 1 (P2T1) treatment was compared to the composite microbial fertilizer treatments (T2) and their utilization practices. Figure 5 shows that during the soybean bloom period, the 0 N fertilizer treatment (P1T3) had slightly smaller soil porosity compared to the Jinyuan 55 conventional fertilizer application (P1T1), with a 16.6% decrease in porosity compared to the T1 treatment (*p* < 0.05). The composite microbial fertilizer had a positive effect on increasing soil porosity during soybean maturity. Compared to the Jinyuan 55 conventional fertilization (P1T1), the soil porosity of the 1/2 nitrogen fertilizer treatment (P1T2) significantly increased, with a 12.03% increase in porosity compared to the T1 treatment (*p* < 0.05). Compared to the conventional fertilizer application (P2T1), neither of the composite microbial fertilizer treatments (T2 and T3) showed significant differences. Among the fertilizer treatments, T3 had the highest total porosity of 56.23% (*p* < 0.05). This indicates that the combined effect of compound microbial fertilizer and chemical fertilizer on soil porosity is more pronounced, while the single application of chemical fertilizer has a lesser impact on soil porosity. The use of compound microbial fertilizer and chemical fertilizer together can significantly increase soil porosity, promoting soil water permeability and air permeability.

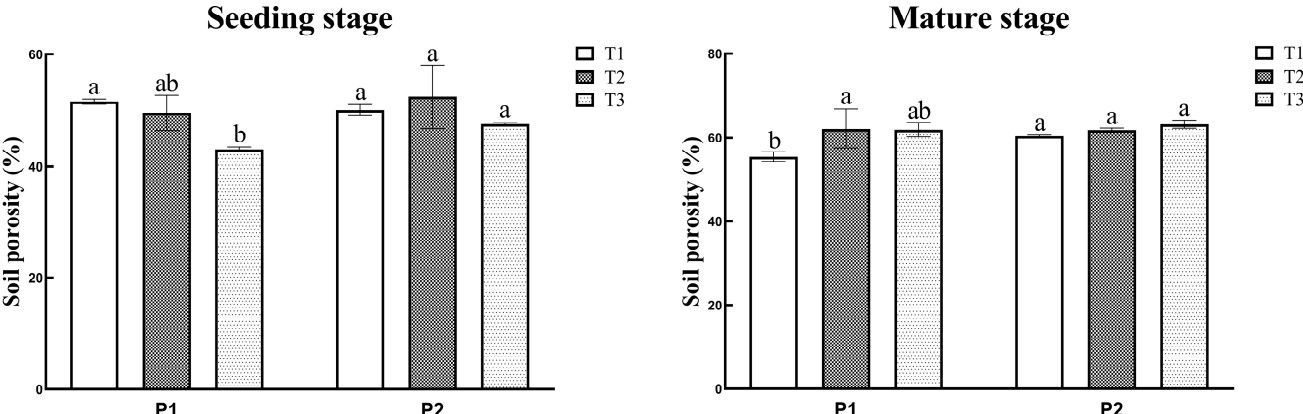

**Figure 5.** Effect of chemical fertilizers with microbial fertilizer complexes on total soil porosity. Note: Different lowercase letters after a number indicate a significant difference (*p* < 0.05).

### 3.1.4. Effect of Compound Microbial Fertilizer with Chemical Fertilizer on Soil Mechanical Composition

The mechanical composition of soil is a fundamental component that greatly influences the physicochemical properties of soil. In this study conducted in 2022, the contents of soil sand, chalky grains, and clayey grains were analyzed in three different fertilizer treatments for two soybean varieties. Figure 6 shows that the soil texture in all six fertilizer treatments was predominantly chalky clay loam, although the composition of the soil varied in terms of grain levels. Table 3 provides a visual representation, indicating that in the case of Jinyuan 55 (P1), the P1T2 treatment had the highest soil sand content. Additionally, both the P1T2 and P1T3 treatments exhibited higher sand content compared to the P1T1 treatment. The P1T1 treatment had a higher powder grain content compared to the P1T2 and P1T3 treatments. There were significant differences in the characteristics of fine sand grains between treatments, with the T2 and T3 treatments having a higher content of sticky grains than the T1 treatment. Specifically, the performances of the P1T2 and P1T3 treatments were 9.96% and 25.90% higher, respectively, than that of the P1T1 treatment. Additionally, the fine sand contents of the P1T2 and P1T3 treatments were 28.64% and 50.51% higher, respectively, than that of the T1 treatment.

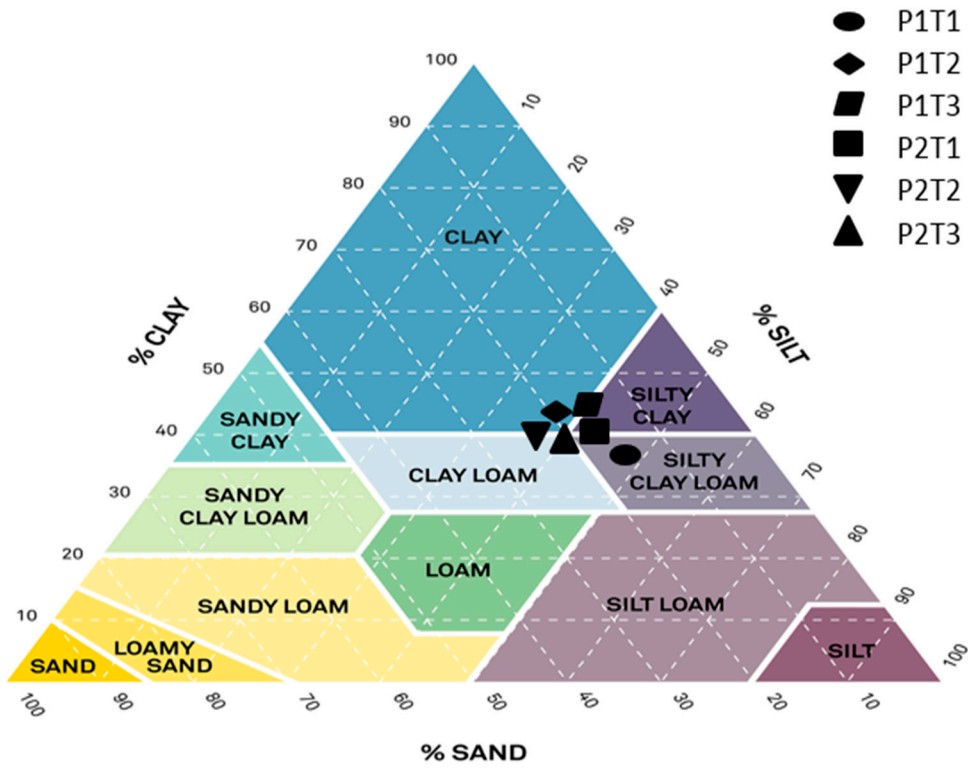

**Figure 6.** Soil texture triangulation.

**Table 3.** Effect of chemical fertilizers with microbial fertilizer complexes on the mechanical composition of soils.

| Process Name | Coarse Middle Sand (%) | Fine Sand (%) | Coarse Silt (%) | Middle Silt (%) | Fine Silt (%) | Clay (%) |
|---|---|---|---|---|---|---|
| P1T1 | 7.20 | 5.90 | 12.28 | 27.43 | 14.45 | 32.74 |
| P1T2 | 9.35 | 7.59 | 8.29 | 26.56 | 11. 62 | 36.60 |
| P1T3 | 7.91 | 8.88 | 8.90 | 22.65 | 10.44 | 41.22 |
| P2T1 | 10.61 | 3.05 | 12.02 | 22.11 | 12.14 | 40.07 |
| P2T2 | 8.54 | 6.58 | 10.31 | 20.30 | 12.04 | 42.22 |
| P2T3 | 15.52 | 4.06 | 9.06 | 19.15 | 10.05 | 42.15 |

In Keshan 1 (P2), the performance among treatments was similar to that of P1. There were significant differences in the amount of fine sand and clay grains between treatments. The P2T2 and P2T3 treatments showed an increase of 5.37% and 5.19% in clay grain content, respectively, compared to the P2T1 treatment. Similarly, the P2T2 and P2T3 treatments showed an increase of 115.74% and 33.11% in fine sand grain content, respectively, compared to the P2T1 treatment.

These findings indicate that the application of compound microbial fertilizer on soil with the same texture at the same experimental site can increase the content of sand and clay particles. Furthermore, the application of compound microbial fertilizer has a positive impact on the generation and storage of coarse and fine sand and clay particles. Additionally, the application of compound microbial fertilizer, in combination with local conventional fertilizer, can increase the content of sand and clay particles in the soil.

### 3.1.5. Effect of Chemical Fertilizers with Complex Microbial Fertilizers on the Composition of Soil Water-Stable Aggregates

The water-stable agglomerate quality of different composite microbial fertilizer treatments was relatively consistent during the maturity period from 2021 to 2022. The results (Figure 7) showed that in 2021, in Jinyuan 55 (P1), the water-stable agglomerate quality of P1T2 and P1T3 treatments was higher than that of P1T1 treatments, but there were no significant differences observed. This suggests that the application of composite microbial fertilizers had a minimal impact on the quality of the water-stable agglomerates. However, the distribution of water-stable cluster mass varied more among different composite microbial fertilizer treatments compared to the characteristics of water-stable cluster mass. Among the different treatments, the proportion of agglomerates ranging from 2 to 5 mm was approximately 11 to 14%, which was the lowest compared to other particle sizes. The proportions of agglomerates ranging from 5 to 10 mm and 1 to 2 mm were 12 to 16% and 15 to 19%, respectively, with no significant difference observed between the treatments. The largest proportion of agglomerates, more than 18%, was found in the 0.5–1 mm size range, specifically in the P1T1 treatment. In contrast, the highest proportion of agglomerates in the P1T3 treatment was observed in the 1–2 mm and 2–5 mm size ranges. In Keshan 1 (P2), the mass of water-stable agglomerates was higher in the P2T3 treatment compared to the P2T2 and P2T1 treatments, although the difference was not statistically significant. Among the treatments, the proportion of agglomerates with particle sizes of 2–5 mm ranged from 11% to 14%. The proportion of agglomerates with particle sizes of 5–10 mm and 1–2 mm ranged from 13% to 20% and 15% to 18%, respectively. The P2T3 treatment had significantly higher proportions compared to the P2T1 treatment. The proportion of agglomerates with particle sizes of 0.5–1 mm was the highest, exceeding 16% in all treatments, with the P2T1 treatment having the highest proportion. The P2T3 treatment had the highest proportion of agglomerates with grain sizes of 5–10 mm, 2–5 mm, and 1–2 mm. The results of the two-year experiment indicated that when the proportion of large-size agglomerates was similar, the application of compound microbial fertilizer promoted the formation of agglomerates in the range of 1–5 mm particle size.

Figure 8 shows that there was no significant difference in the mean weight diameter (MWD) of soil water-stable aggregates in soybean fields when chemical fertilizer with compound microbial fertilizer was used. In 2021, for Jinyuan 55 (P1), the MWD of the T2 treatment was lower than that of the T1 treatment, while the MWD of the T1 and T3 treatments remained the same. Similarly, in Keshan 1 (P2), the MWD of the T2 and T3 treatments increased compared to the T1 treatment, but the difference between treatments was not significant. In 2022, the MWD trend for the three treatments of Jinyuan 55 (P1) and Keshan 1 (P2) was similar, with the MWD of the T2 treatment being slightly lower than that of the T1 treatment, and the MWD of the T3 treatment being higher than that of the T1 treatment. However, the difference was not significant.

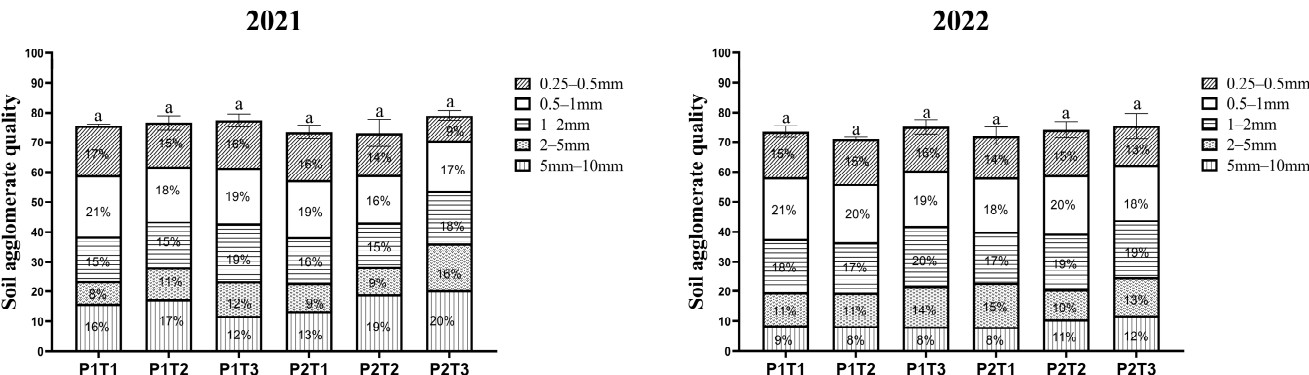

**Figure 7.** Effect of chemical fertilizers with microbial fertilizer complexes on the composition of soil water-stable agglomerates. Note: Different lowercase letters after a number indicate a significant difference ($p < 0.05$).

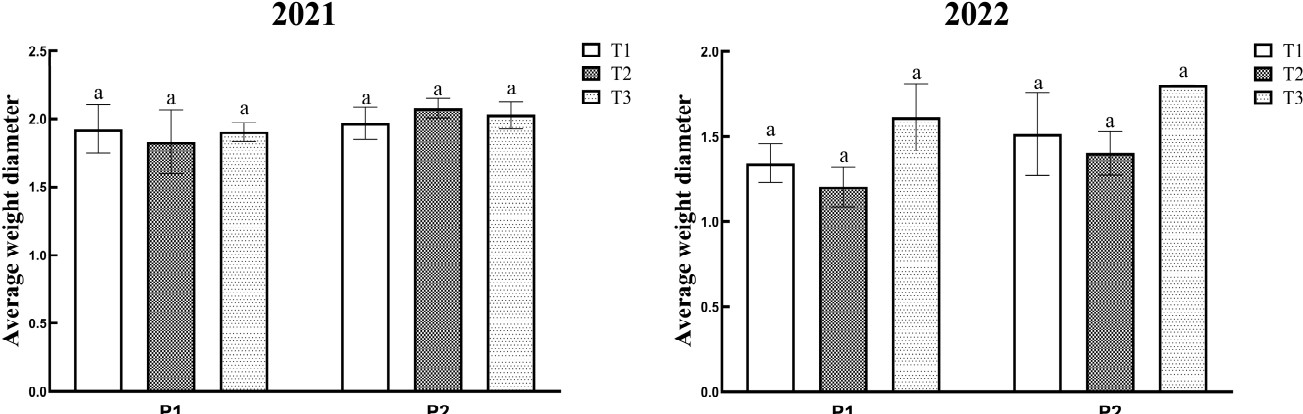

**Figure 8.** Effect of chemical fertilizers with microbial fertilizer complexes on the mean weight diameter of soil water-stable agglomerates. Note: Different lowercase letters after a number indicate a significant difference ($p < 0.05$).

According to Figure 9, in 2021, the T2 and T3 treatments had slightly higher geometric mean diameter (GMD) compared to the T1 treatment in Jinyuan 55 (P1), but the differences between treatments were not significant. In Keshan 1 (P2), the GMD of the T2 treatment was significantly lower than that of the T1 treatment, while the GMD of the T3 treatment was similar to that of the T1 treatment, but the differences among treatments were not significant. In 2022, in Jinyuan 55 (P1), the GMD of the T3 treatment was slightly higher than that of the T1 treatment, and the GMD of the T2 treatment was slightly lower than that of the T1 treatment, but the differences were not significant. In Keshan 1 (P2), it was observed that the GMD of the T3 treatment was higher than that of the T1 treatment, while the GMD of the T1 and T2 treatments was similar, and the differences among treatments were not significant.

The application of compound microbial fertilizers helps increase the stability of water-stable aggregates. The use of a larger quantity of compound microbial fertilizers (T3) has a greater impact, effectively protecting soil aggregates in the T3 treatment and speeding up their reorganization and recovery after damage to the soil water-stable aggregates.

*3.2. Effect of Chemical Fertilizer with Compound Microbial Fertilizer on Soybean Yield*

The agronomic characteristics of soybeans are influenced by various fertilizer application methods, which result in yield fluctuations. In 2021, extreme precipitation conditions affected the yields of different soybean varieties in each treatment, as well as their yield components, as demonstrated in Table 4.

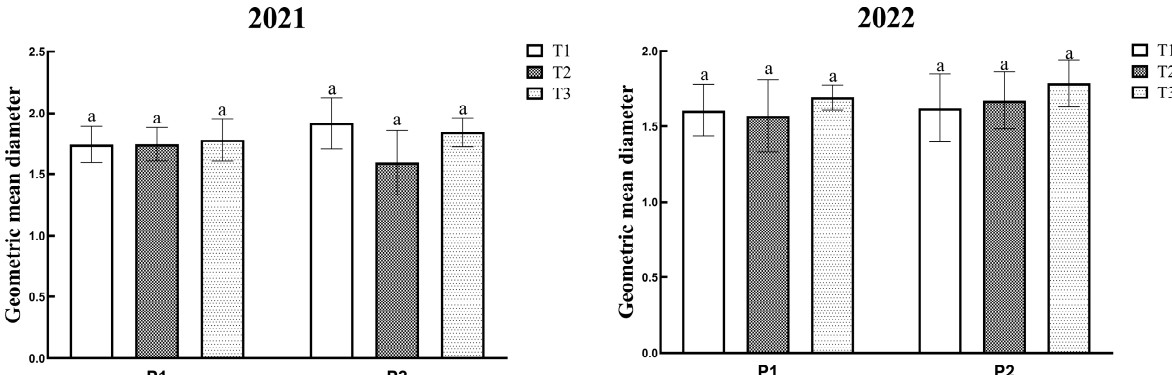

**Figure 9.** Effect of chemical fertilizers with microbial fertilizer complexes on the geometric mean diameter of soil water-stable agglomerates. Note: Different lowercase letters after a number indicate a significant difference ($p < 0.05$).

**Table 4.** Effect of chemical fertilizer with microbial fertilizer complex on soybean yield and yield components. Note: Different lowercase letters after a number indicate a significant difference ($p < 0.05$).

| Years | Varieties | Treat Menu | Pod Number | Seed Number | 100 Seed Weight/g | Seed Weight/g | Yield/kg·km$^{-2}$ |
|---|---|---|---|---|---|---|---|
| 2021 | Jinyuan 55 | P1T1 | 24.07 ± 3.99 a | 54.58 ± 8.35 a | 18.83 ± 0.01 a | 9.78 ± 0.57 ab | 3014.94 ± 55.06 a |
| | | P1T2 | 26.00 ± 2.42 a | 57.23 ± 3.78 a | 19.78 ± 1.09 a | 10.91 ± 0.94 a | 3407.44 ± 183.44 a |
| | | P1T3 | 20.33 ± 6.97 a | 43.27 ± 3.55 a | 18.33 ± 0.49 a | 7.78 ± 0.59 b | 2733.38 ± 104.00 a |
| | Keshan 1 | P2T1 | 21.03 ± 4.64 a | 51.01 ± 6.41 a | 18.53 ± 2.60 a | 8.67 ± 1.95 a | 3195.49 ± 89.26 a |
| | | P2T2 | 27.87 ± 4.53 a | 52.30 ± 7.48 a | 19.97 ± 0.33 a | 9.18 ± 2.02 a | 3366.25 ± 250.14 a |
| | | P2T3 | 20.46 ± 2.21 a | 46.50 ± 6.84 a | 19.19 ± 0.74 a | 8.06 ± 0.13 a | 2899.56 ± 245.62 a |
| 2022 | Jinyuan 55 | P1T1 | 29.13 ± 1.44 a | 71.66 ± 3.91 ab | 19.66 ± 0.21 a | 12.72 ± 0.56 ab | 3787.47 ± 93.76 b |
| | | P1T2 | 31.15 ± 1.35 a | 76.48 ± 8.15 a | 20.68 ± 0.26 a | 14.92 ± 0.91 a | 4264.60 ± 73.60 a |
| | | P1T3 | 23.54 ± 0.74 b | 54.23 ± 5.17 b | 20.13 ± 0.13 a | 10.81 ± 1.04 b | 3135.03 ± 101.39 c |
| | Keshan 1 | P2T1 | 29.10 ± 5.04 a | 74.13 ± 16.1 a | 19.81 ± 0.71 a | 13.15 ± 0.73 a | 3526.80 ± 328.00 ab |
| | | P2T2 | 34.40 ± 5.16 a | 78.83 ± 5.33 a | 20.21 ± 0.16 a | 14.75 ± 1.47 a | 4247.33 ± 91.66 a |
| | | P2T3 | 24.83 ± 2.85 a | 56.29 ± 3.44 a | 18.19 ± 0.09 b | 10.59 ± 1.24 a | 3124.30 ± 269.00 b |

Compared to the conventional fertilization method Jinyuan 55 (P1T1), the treatment P1T2 (50% nitrogen fertilizer + compound microbial fertilizer) showed a certain increase in the number of pods per plant, 100 grain weight, grain weight per plant, and yield. The difference in grain weight per plant between P1T2 and P1T1 was significant, and the soybean yield in P1T2 increased by 13.02% compared to P1T1 ($p < 0.05$). However, the treatment P1T3 (0 nitrogen fertilizer + compound microbial fertilizer) resulted in a 9.34% lower soybean yield compared to P1T1. In comparison to the conventional fertilization method Keshan 1 (P2T1), the treatment P2T2 (50% nitrogen fertilizer + compound microbial fertilizer) showed some improvement in the number of pods per plant, number of grains per plant, 100 grain weight, grain weight per plant, and yield. However, these improvements did not reach a significant level. The yield of soybean in the P2T2 treatment increased by 5.34% compared to the P2T1 treatment, while the yield of soybean in the P2T3 treatment decreased by 9.26% compared to the P2T1 treatment.

The yields in all varieties of treatments increased in 2022 compared to 2021. The P1T2 treatment of Jinyuan 55 (50% nitrogen + microbial fertilizer) exhibited higher numbers of pods per plant, 100 kernel weight, kernel weight per plant, and overall yield compared to the conventional fertilizer (P1T1). The soybean yield in P1T2 was significantly higher by 12.6% ($p < 0.05$) than in P1T1. However, the number of pods per plant, 100 seed weight, grain weight per plant, and yield were lower in the P1T3 treatment (0 nitrogen fertilizer + complex microbial fertilizer) compared to P1T1. Soybean yield in P1T3 was significantly reduced by 17.21% ($p < 0.05$) compared to P1T1. As for Keshan 1, the P2T2 treatment (50% nitrogen + microbial fertilizer complex) increased the number of pods per plant, number of grains per

plant, 100 grain weight, grain weight per plant, and total soybean yield compared to the conventional fertilizer (P2T1). Yield increased by 20.4%. The study examined the effects of a combination of nitrogen and compound microbial fertilizers on soybean yield. The results indicated that when compared to plants treated solely with nitrogen fertilizer, the plants treated with the combination fertilizer exhibited a decrease in the number of pods and grains per plant, 100 grain weight, single grain weight, and overall yield. Notably, the P2T3 treatment showed a significant 8.18% reduction in 100 grain weight compared to P2T1. Furthermore, the soybean yield of plants treated with P2T3 decreased by 11.41% in comparison to P2T1.

The study also found that using compound microbial fertilizer in moderation can have a positive impact on soybean yield over two years. However, excessive use of fertilizer negatively affects the yield.

### 3.3. Correlation Analysis

The results of correlation analysis showed that the yield had a highly significant negative correlation with MWD ($p < 0.001$; Figure 10), and a highly significant positive correlation with 100 grain weight, the number of grains per plant, and grain weight per plant ($p < 0.001$). The yield was also highly significantly and positively correlated with seedling soil bulk weight, seedling soil porosity, and 100 seed weight ($p < 0.01$), and significantly and negatively correlated with GMD ($p < 0.05$). Seedling soil porosity exhibited a highly significant negative correlation with maturity soil porosity ($p < 0.001$). Soil porosity at maturity was highly significant and negatively correlated with soil water content ($p < 0.001$). MWD showed a highly significant negative correlation with pod number, seed number, and seed weight ($p < 0.001$). Pod number was highly significantly positively correlated with seed number and seed weight ($p < 0.001$). Seed number was highly significantly and positively correlated with 100 seed weight and seed weight ($p < 0.001$). The 100 seed weight was highly significantly and positively correlated with seed weight ($p < 0.001$). GMD was highly significantly and negatively correlated with seed number and seed weight ($p < 0.01$). The pod number was highly significantly and positively correlated with 100 seed weight.

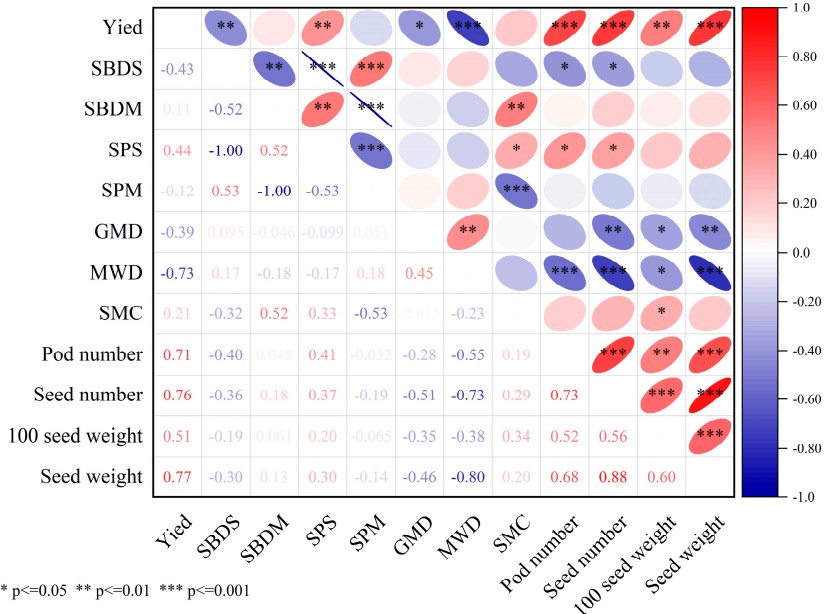

**Figure 10.** Analysis of the correlation between indicators of chemical fertilizers with microbial compound fertilizers. Note: Soil bulk density at seedling stage (SBDS), soil bulk density at maturity (SBDM), soil porosity at seedling stage (SPS), soil porosity at maturity (SPM), geometric mean diameter (GMD), mean weight diameter (MWD), soil moisture content (SMC), *, ** and *** indicate the significant effects of farming practices at $p < 0.05$, $p < 0.01$, and $p < 0.001$, respectively, determined by Pearson's correlation.

## 4. Discussion

### 4.1. Effect of Chemical Fertilizer with Compound Microbial Fertilizer on Soil Physical Properties

Fertilizer application is an important factor affecting the physical properties of soil, and long-term over-application of chemical fertilizers can adversely affect soil physico-chemical properties [9]. In this study, after 2 years of continuous application of compound microbial fertilizer for planting, the beneficial microorganisms in compound microbial fertilizer (such as nitrogen-fixing bacteria and phosphorus-solubilizing bacteria) were able to decompose the organic matter, promote the agglomeration and cementation of the soil particles, and form a good soil structure. This improves soil porosity and permeability and increases soil aeration and water retention capacity [22]. The results of this study showed that both (0 nitrogen fertilizer + compound microbial fertilizer) T3 and (50% nitrogen fertilizer + compound microbial fertilizer) T2 reduced the soil capacity compared to (chemical fertilizer alone) T1. Zhang Xiting et al. [23] stated that the optimum soil capacity interval for soybean growth is between 1.00 and 1.30 g/cm$^3$. Therefore, the soil capacity of the single application of compound microbial fertilizer and the alternative treatment of compound microbial fertilizer in this experiment meets the requirements for soybean growth. The results of this experiment showed that the use of compound microbial fertilizer for two consecutive years could effectively reduce the soil bulk weight and increase the total soil porosity. Compared with no compound microbial fertilizer T1 (CK), the compound microbial fertilizer treatments (T2 and T3) reduced the soil bulk weight and increased the soil porosity, with the best effect in the (0 nitrogen fertilizer + compound microbial fertilizer) T3 treatment. In the present study, microorganisms in the composite microbial fertilizer helped the soil maintain moderate moisture content and increased the water-holding capacity of the soil. They are able to decompose organic matter to release extracellular polysaccharides and form sticky substances that help to keep water available in the soil and reduce water evaporation and loss [24]. Soil particle size composition is one of the most important physical properties of soil, which has an important effect on soil moisture and fertility storage and is closely related to soil structural stability [25]. The microorganisms in the compound microbial fertilizer produce extracellular polysaccharide substances, which contribute to the agglomeration of clayey soil and make it easier to loosen. At the same time, microorganisms can also decompose the sticky particles and colloids in the soil, reducing its viscosity and improving its texture [26,27]. In addition, the soil formation conditions of the treatments in this paper were basically the same, and the soybean planting and field management methods were also the same. The only difference was the amount of compound microbial fertilizer applied. Therefore, fertilizer application was the main factor influencing the change in soil particle size distribution in this experiment. This study showed that in the 0–20 cm soil layer, the microorganisms in the composite microbial fertilizer produced extracellular acids through metabolic processes that reacted with soil minerals and contributed to the stabilization of soil aggregates. By increasing the stability of soil aggregates, soil compaction and topsoil loading can be reduced, and soil permeability can be improved [9]. The distribution of soil granular components changed with the application of treatments of complex microbial fertilizers (T2 and T3) compared to the application of chemical fertilizer T1 alone. Application of chemical fertilizer T1 alone increased the volume fraction of clay and chalk and decreased the volume fraction of sand particles, whereas organic and inorganic treatments decreased the volume fraction of clay particles and increased the volume fraction of sand particles. However, the treatment with (0 nitrogen fertilizer + composite microbial fertilizers) T3 gave the best results. Fertilizer application changed the composition of soil particles compared to T1 application, especially with the addition of composite microbial fertilizer or only composite microbial fertilizer, which was more effective and favored the stabilization of soil structure. Soil aggregates are clumps of soil that are cemented into different forms [28], which are the main units of soil structure [29], and the distribution of different particle sizes of aggregates is closely related to the stability of soil structure [30]. In this experiment, the application of compound microbial fertilizer treatments (T2 and T3) increased the

content of large particle-size agglomerates in the soil, with the most obvious effect being the (0 nitrogen fertilizer + complex microbial fertilizer) T3 treatment, which was consistent with the results of Zhou Zhengxiong et al. [31]. In a previous study of this experimental field, the soil was divided into >2 mm, 0.25–2 mm, 0.053–0.25 mm, and <0.053 mm soil particle sizes by the wet sieving method, and it was concluded that the application of composite microbial fertilizer could significantly increase the agglomerates with particle sizes >2 mm and reduce the agglomerates with particle sizes 0.25–2 mm. The results of the current experiment showed that in the 0–20 cm soil level, all treatments had the highest content of >5 mm agglomerates, and compared with the application of chemical fertilizer alone (T1), the application of composite microbial fertilizer treatments (T2 and T3) promoted the formation of large soil agglomerates; they all reduced micro-agglomerates <0.25 mm, which was in agreement with the results of the previous study [32–34]. The reason may be that the application of compound microbial fertilizer increased the soil organic matter and humus content of the soil, which made the soil particles easy to cement and thus accelerated the formation of soil agglomerates [35,36]. Agglomerate stability is usually assessed by a combination of mean weight diameter (MWD), geometric mean weight diameter (GMD), and other indicators [37]. In the present study, it was found that the application of chemical fertilizer alone disturbed soil structure at the 0–20 cm level, whereas the application of composite microbial fertilizer increased soil structural stability. Complex microbial fertilizers at different use levels significantly increased MWD and GWD compared to chemical fertilizer alone T1 treatment, with (0 nitrogen fertilizer + complex microbial fertilizer) T3 treatment exhibiting the greatest increase.

### 4.2. Effect of Chemical Fertilizer with Compound Microbial Fertilizer on Soybean Yield Formation

Microorganisms in compound microbial fertilizers can produce organic acids, enzymes, and hormones through metabolic activities to regulate the soil environment [14] and promote the reproduction and activities of soil microorganisms. These microorganisms can form a symbiotic relationship with the plant root system and enhance the nutrient uptake capacity of crops [38]. In this study, the combination of chemical fertilizer with different amounts of compound microbial fertilizer was effective in increasing soybean yield, similar to previous results [39,40], mainly because the combination of chemical fertilizer and compound microbial fertilizer could better release organic acids [41], increase the number of microorganisms and microbial activity, and make the fertilizer and water more readily available for uptake and utilization [42]. The combination is more capable of meeting the nutritional needs of soybeans at all times, thus making it easier to increase soybean yields, while the microorganisms in the compound microbial fertilizer are able to decompose organic matter and release nutrient elements to increase soil fertility. The decomposition of the organic matter also generates heat, which raises the soil temperature and is conducive to the growth of the crop [43]. Compared with the use of conventional fertilizers and naturally grown soybeans, legume crops applied with microbial fertilizers have a certain degree of improvement in plant height, pod number, 100 grain weight, and other indicators [44]. Liang Xiaotian et al. [45] studied the effect of microbial foliar fertilizers on soybeans and found that most of the microbial foliar fertilizers increased the 100 grain weight of soybean by about 4% and the grain weight per plant by about 10%. Zhao Nianli et al. [46] found that the 100 grain weight of soybeans increased by about 13% or more at the late stage of development when soybeans were treated with a high-yielding Russian rhizobial fertilizer. Ren Tinghu et al. [47] showed that 50% organic and 50% inorganic N management resulted in the most significant increase in yield and efficiency, which is consistent with the results of this experiment. The fertilization effect of compound microbial fertilizers varied from trial to trial due to differences in cultivated crop species, type and amount of compound microbial fertilizers added, as well as trial location and climate [48]. Meanwhile, in this study, it was found through field experiments that chemical fertilizer with compound microbial fertilizer promoted yield increases, mainly caused by the increase in the number of grains per plant, the number of pods per plant, and

the grain weight per plant. Therefore, increasing or maintaining a higher number of pods per plant and grains per plant is the key to increasing soybean yield, which is consistent with the study of Li Guoqing [49].

## 5. Conclusions

The two-year experiment demonstrated that the application of complex microbial fertilizers (T2 and T3) had a more positive impact on the soil. It reduced soil bulk weight, increased soil porosity, and enhanced the number of soil aggregates. Additionally, the composite microbial fertilizer treatments (T2 and T3) resulted in a decrease in the volume fraction of clay particles and an increase in the volume fraction of sand particles, compared to the conventional fertilizer application without composite microbial fertilizer (T1) treatment. Notably, the best results were achieved with the composite microbial fertilizer treatment T3 alone. The application of fertilizers also altered the soil particle composition, and the composite microbial fertilizer treatments (T2 and T3) played a crucial role in stabilizing the soil structure when compared to the T1 treatment without composite microbial fertilizer.

T3 (0 nitrogen fertilizer + complex microbial fertilizer) exhibited a decrease in yield compared to T1 (conventional fertilizer application without complex microbial fertilizer). When chemical fertilizer and compound microbial fertilizer were combined, the optimal amount of compound microbial fertilizer was found to be 2.023 t-hm$^{-2}$, resulting in a theoretical maximum soybean yield of 4264.60 kg/hm$^{-2}$. However, the yield decreased instead of increasing when the amount of compound microbial fertilizer applied exceeded 2.023 t-hm$^{-2}$.

According to this study, it is recommended to apply compound microbial fertilizer at a rate of 2.023 t-hm$^{-2}$ along with a 50% reduction in nitrogen fertilizer. This application can help protect the black soil, enhance its self-healing ability, improve the quality of soybeans, and contribute to the development of a green fertilizer system for the soybean industry. The findings of this research provide a valuable practical foundation for the construction and development of a green fertilizer system in the global soybean industry.

**Author Contributions:** Data curation, M.W.; Formal analysis, X.J.; Investigation, Y.Z.; Software, B.Q.; Validation, W.M.; Writing—original draft, C.F. All authors have read and agreed to the published version of the manuscript.

**Funding:** This study was supported by the National Key Research and Development Programme of China (2022YFD1500105); the cooperative project of Jiusan Soybean Industry Innovation Research Institute (2022101); and the postgraduate research and innovation funding project of Heilongjiang Bayi Agricultural and Reclamation University (YJSCX2021-NXY10).

**Data Availability Statement:** Data are present in the article.

**Conflicts of Interest:** The authors declare no conflict of interest.

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
