# Peer review of "Effect of Chemical Fertilizer with Compound Microbial Fertilizer on Soil Physical Properties and Soybean Yield"

_agronomy, doi:10.3390/agronomy13102488_

Round 1

Reviewer 1 Report

The study presented here explores the potential benefits of compound microbial fertilizer as an alternative to chemical fertilizer for soil improvement. While the research findings do suggest some positive impacts on soil physical properties and soybean yield, some several critical points and limitations should be addressed:

1.      The study compares compound microbial fertilizer (T2 and T3) with conventional chemical fertilizer (T1). However, it would be more informative if the research included a control group with no fertilizer application to highlight the true impact of compound microbial fertilizer alone.

2.      The text is filled with extensive data and references to prior studies, but it lacks clarity regarding data interpretation. Readers may need help understanding the significance of the results presented.

3.      The study needs to provide more details on the specific composition of the compound microbial fertilizer used. With this information, it's easier for other researchers to replicate the experiments and determine whether similar results can be achieved with different formulations.

4.      Soil properties vary significantly by region, climate, and crop type. The study's conclusions may not apply universally to a specific experimental field. More information is needed on the soil characteristics and geographic location to assess the generalizability of the findings.

5.      While the text mentions differences in soil properties and crop yield between treatments, it is essential to include statistical analysis to determine whether these differences are statistically significant. Without statistical significance, it is challenging to draw firm conclusions.

6.      The study suggests that compound microbial fertilizer enhances microbial activity but lacks a detailed analysis of how this affects soil microbiota. Understanding the shifts in microbial communities can provide valuable insights into the soil health implications.

7.      There is no discussion of potential environmental impacts, such as the release of excess nutrients into surrounding ecosystems or the long-term sustainability of using compound microbial fertilizers.

Overall, the study highlights some potential benefits of using compound microbial fertilizer, There are several limitations and areas for improvement need to be addressed. Future research should aim for greater clarity, statistical rigor, and consideration of adopting such fertilizers' broader ecological and economic implications.

 Moderate editing of English language required

Author Response

The study presented here explores the potential benefits of compound microbial fertilizer as an alternative to chemical fertilizer for soil improvement. While the research findings do suggest some positive impacts on soil physical properties and soybean yield, some several critical points and limitations should be addressed:

  1. The study compares compound microbial fertilizer (T2 and T3) with conventional chemical fertilizer (T1). However, it would be more informative if the research included a control group with no fertilizer application to highlight the true impact of compound microbial fertilizer alone.

Answer: Research on the effects of microbial fertilizers alone on crop growth and soil has been extensive and has yielded good results, however, the use of microbial fertilizers alone is economically expensive and the nutrient composition is difficult to maintain for subsequent crop growth and yield formation. In order to highlight the effects of microbial fertilizers in combination with chemical fertilizers, we have used a combination of the two, focusing on the efficient use of nutrients and yield formation of the crop as a result of the combination. We will continue our research in this direction in subsequent production experiments, and conduct comparative analyses with individual fertilizer applications. We are grateful to the reviewers for their valuable comments and will continue to monitor the situation.

  1. The text is filled with extensive data and references to prior studies, but it lacks clarity regarding data interpretation. Readers may need help understanding the significance of the results presented.

Answer: A correlation analysis has been added to the discussion section to strengthen the links between the data. The discussion section has been elaborated and explained to highlight the interpretation of the study data to the final results. Thank you to the reviewers for their valuable comments, which have improved my article.

  1. The study needs to provide more details on the specific composition of the compound microbial fertilizer used. With this information, it's easier for other researchers to replicate the experiments and determine whether similar results can be achieved with different formulations.

Answer: The compound microbial fertilizer used was provided by Zhaofeng Hemei Company. It had an organic matter content of ≥20%, an effective live bacterial count of ≥0.2 billion/g, Amino acid content of ≥5 %, a sulfur content of ≥10 %, a humic acid content of≥4 %, and a total nutrient content of (N+P2O5+K2O=8%, with a ratio of 2:2:1), It contains nitrogen-fixing bacteria, putrefactive bacteria, phosphorus solubilizing bacteria, tufted mycorrhizal fungi, zygomycetes, etc. I would like to thank the reviewers for their valuable input, which has improved my article.

  1. Soil properties vary significantly by region, climate, and crop type. The study's conclusions may not apply universally to a specific experimental field. More information is needed on the soil characteristics and geographic location to assess the generalizability of the findings.

A: We have provided the relevant soil and two years of meteorological data for the area in the experimental design section to ensure that the experiment will provide readers with a wide range of references in subsequent future studies.

  1. While the text mentions differences in soil properties and crop yield between treatments, it is essential to include statistical analysis to determine whether these differences are statistically significant. Without statistical significance, it is challenging to draw

Answer: statistical analysis of correlation has been added. Thank you to the reviewers for their valuable comments, which have improved my article.

  1. The study suggests that compound microbial fertilizer enhances microbial activity but lacks a detailed analysis of how this affects soil microbiota. Understanding the shifts in microbial communities can provide valuable insights into the soil health implications.

Answer: I have already explained the effects caused by soil microorganisms in the discussion. In the subsequent production experiments we will carry out research in the direction related to soil microorganisms. Thank you again for the valuable comments provided by the reviewers, we will continue to pay attention.

  1. There is no discussion of potential environmental impacts, such as the release of excess nutrients into surrounding ecosystems or the long-term sustainability of using compound microbial fertilizers.

Answer: It has deepened the rich discussion, and this experiment is still continuing in my team. We will pay attention to the study of the above issues in the subsequent production experiments. Thanks to the reviewers for their valuable comments.

Overall, the study highlights some potential benefits of using compound microbial fertilizer, There are several limitations and areas for improvement need to be addressed. Future research should aim for greater clarity, statistical rigor, and consideration of adopting such fertilizers' broader ecological and economic implications.

Answer: I am very grateful to the reviewers for their comments and help, I have benefited a lot. I will continue to improve and revise the experimental design and the paper. Thank you again for your strict requirements.

Reviewer 2 Report

1.     Figure 2 needs p value (statistical significance)

2.     You have taken 3 treatment and 2 varieties. same 3 treatments used for each crop. In this case, the proper statistical design should be Split-plot not CRD.

3.     How may replicates you have taken in your field experiment for 3 treatment and 2 varieties.

4.     What is the total analysis of compound microbial fertilizer? just live bacterial count or NPK(??) analysis will not be sufficient. you should give the type of microbe present like N-fixer, P,K solubilizer etc.      Otherwise this manuscript is just a marketing of one company's product.

6.     Author should check the English grammar.

 Moderate editing of English language required

Author Response

Comments and Suggestions for Authors

  1. Figure 2 needs p-value (statistical significance)

A: It has been added. I would like to thank the reviewers for their valuable input, which has improved my article.

  1. You have taken 3 treatment and 2 varieties. same 3 treatments used for each crop. In this case, the proper statistical design should be Split-plot not CRD.

Answer: Dear reviewer, I have rewritten the experimental design.The trial followed a one-way completely randomized block design with three treat-ments of two varieties. Each treatment was replicated four times on a total of 24 plots. Thank you for your help and advice so that I can perfect my thesis.

  1. How may replicates you have taken in your field experiment for 3 treatment and 2 varieties.

Answer: has been added to the paper with four replications. In order to ensure the breadth and accuracy of the experiment this trial was designed as a completely randomised block group trial.

  1. What is the total analysis of compound microbial fertilizer? just live bacterial count or NPK(??) analysis will not be sufficient. you should give the type of microbe present like N-fixer, P, K solubilizer, etc. Otherwise, this manuscript is just a marketing of one company's product.

Answer: The compound microbial fertilizer used was provided by Zhaofeng Hemei Company. It had an organic matter content of ≥20%, an effective live bacterial count of ≥0.2 billion/g, Amino acid content of ≥5 %, a sulfur content of ≥10 %, a humic acid content of≥4 %, and a total nutrient content of (N+P2O5+K2O=8%, with a ratio of 2:2:1), It contains nitrogen-fixing bacteria, putrefactive bacteria, phosphorus solubilizing bacteria, tufted mycorrhizal fungi, zygomycetes, etc. I would like to thank the reviewers for their valuable input, which has improved my article.

  1. The author should check the English grammar.

Answer: checked, thanks to the reviewers for their valuable comments, which helped me a lot with my article.

Reviewer 3 Report

Dear Authors,

I find the research presented in the manuscript quite interesting and likely to have significance for the development of agronomy. However, as far as the manuscript is concerned, I have quite a few comments that the Authors should address. First of all, putting myself in the role of a reader who does not specialize in this type of research, I would expect the manuscript to answer the question of why the analyzed factors of the field experiment affected soil properties and yield, however, such information is missing from the manuscript.

Abstract: In my opinion, the abstract is too long. I suggest describing in less detail the results obtained. And add one sentence of summary of the results of the conducted experiment.

Introduction: Despite the fact that the conducted field research is about soybeans in the introduction which by definition is supposed to be an introduction to the research there is no mention of soybeans. The economic importance of soybeans, the response of soybeans to the fertilization applied in the field experiment and the benefits of soybeans as a legume should be supplemented. If there is a lack of research on soybeans in the literature I suggest referring to other legumes. This chapter also lacks a specific purpose for conducting the study

Materials and Methods:

L87 In my opinion, accurate weather data should be provided during the implementation of the field experiment

L89 What was the forecrop for the main crop?

L100 For what reason were these varieties chosen? Are these varieties widely grown in the area or are they recent introductions? I also did not find information regarding inoculation of soybean seeds. Has it been used? Inoculation of legumes is a very important issue in their cultivation.

L105 What kind of bacteria was it? This is one (in my opinion) of the main interesting aspects of this experiment. The use of bacteria is a very interesting method that has been studied more and more extensively in studies around the world in various plants. So this is an aspect that is relevant to this experiment

L107 With what machines?

L115 Why is there such a discrepancy in plant density? It was not stated what method of sowing was done, is it related to this? Quite a large discrepancy may have affected the analyzed yield of these plants.

Results: I don't quite understand an issue related to the results of the field tests. Some of the results were discussed in years (soybean yield) and some only in 2022 what is the reason for this? This should be clarified and clearly stated in the manuscript. In addition, in this section there are statements that in my opinion should be in the discussion section

L337 Weather conditions were mentioned without any information about them. As I stated earlier I believe that they should be stated

Discussion

L380 In this subsection, except for one sentence, there is no description/explanation of the mechanisms that determined why the variation of fertilization between chemical and microbial influences the physical properties of the soil. When I read the manuscript, I was hoping for a broad description of the influence of the field experiment factors on the results obtained, but it is missing here. If the manuscript were to be published it is imperative that this be explained and supplemented. This is one of the key aspects for this manuscript.

L443 What are the findings?

L445 The reference to the literature must be added

L451 Why did this situation occur? What mechanisms determined this?

L462 Another aspect in favor of adding the weather results of the experiment. But there is also another question why microbial fertilizers affect plants due to different factors

L466 But why was this increased with a given fertilization?

In addition, several Figures are very unreadable. The authors mentioned that they were prepared in MWord, I suggest improving them. Maybe add colors and increase their size?

The References section is not uniform and not fully prepared properly, as are the references in the manuscript. I suggest you read the requirements for Authors and correct this.

Author Response

I find the research presented in the manuscript quite interesting and likely to have significance for the development of agronomy. However, as far as the manuscript is concerned, I have quite a few comments that the Authors should address. First of all, putting myself in the role of a reader who does not specialize in this type of research, I would expect the manuscript to answer the question of why the analyzed factors of the field experiment affected soil properties and yield, however, such information is missing from the manuscript.

Answer: I have added heat maps of correlation between yield components and soil physical properties. And enrich the related research in the discussion to increase the rationality of the article. We will continue to keep the research in the related direction in the subsequent production experiments. Thanks again to the reviewers for their valuable comments, and we will continue to pay attention to it.

Abstract: In my opinion, the abstract is too long. I suggest describing in less detail the results obtained. And add one sentence of summary of the results of the conducted experiment.

Answer: I have revised the abstract section by summarising and summarising it to highlight the research highlights and improve the readability for the authors. Thank you to the reviewers for providing valuable comments, which have improved my article.

Introduction: Despite the fact that the conducted field research is about soybeans in the introduction which by definition is supposed to be an introduction to the research there is no mention of soybeans. The economic importance of soybeans, the response of soybeans to the fertilization applied in the field experiment and the benefits of soybeans as a legume should be supplemented. If there is a lack of research on soybeans in the literature I suggest referring to other legumes. This chapter also lacks a specific purpose for conducting the study

Answer: Thanks to the editor's valuable comments, I have given a brief introduction to the economic importance of soybeans in the preface section and have revised the whole to ensure consistency. Once again, I thank the reviewers for their valuable comments, which have greatly improved my article.

Materials and Methods:

L87 In my opinion, accurate weather data should be provided during the implementation of the field experiment

Answer: Thanks to the editor's valuable comments, I have added meteorological data related to crop fertility to the overview of the experimental site.

L89 What was the forecrop for the main crop?

Answer: In order to avoid the hazards of continuous cropping, the previous crop in this trial was maize and no other experimental design was carried out on the previous crop in order to ensure the stability of the soil in the individual plots.

L100 For what reason were these varieties chosen? Are these varieties widely grown in the area or are they recent introductions? I also did not find information regarding inoculation of soybean seeds. Has it been used? Inoculation of legumes is a very important issue in their cultivation.

Answer: Dear reviewer, thank you very much for your comments. The two varieties I chose, P1 is a high-protein variety mainly cultivated locally, inoculated and identified as moderately resistant to mosaic virus disease strain 1, susceptible to mosaic virus disease strain 3, and moderately susceptible to grey spot disease. The crude protein content was 42.19% and crude fat content was 19.60%. P2 is a high-fat variety mainly cultivated in the region and was identified by inoculation as moderately susceptible to grey spot, moderately susceptible to mosaic virus disease No. 1 strain, and susceptible to mosaic virus disease No. 3 strain. The crude protein content was 38.04% and crude fat content was 21.82%. The two varieties of soybeans were selected for this trial because field management was consistent with locally grown production. There are precedents for their use in the following papers.

Binbin Q, Weixin Z, Xingjie Z, et al. Effect of nitrogen application levels on photosynthetic nitrogen distribution and use efficiency in soybean seedling leaves.[J]. Journal of plant physiology, 2023,287. DOI:10.1016/J.JPLPH.2023.154051

L105 What kind of bacteria was it? This is one (in my opinion) of the main interesting aspects of this experiment. The use of bacteria is a very interesting method that has been studied more and more extensively in studies around the world in various plants. So this is an aspect that is relevant to this experiment

Answer: The compound microbial fertilizer used was provided by Zhaofeng Hemei Company. It had an organic matter content of ≥20%, an effective live bacterial count of ≥0.2 billion/g, Amino acid content of ≥5 %, a sulfur content of ≥10 %, a humic acid content of≥4 %, and a total nutrient content of (N+P2O5+K2O=8%, with a ratio of 2:2:1), It contains nitrogen-fixing bacteria, putrefactive bacteria, phosphorus solubilizing bacteria, tufted mycorrhizal fungi, zygomycetes, etc. I would like to thank the reviewers for their valuable inputs, which have improved my article.

L107 With what machines?

Answer: a brand of seeder in China, transformed by myself in order to integrate open furrow seeder. Thank you for the teacher's attention.

L115 Why is there such a discrepancy in plant density? It was not stated what method of sowing was done, is it related to this? Quite a large discrepancy may have affected the analyzed yield of these plants.

Answer: Dear reviewer, Thank you for your findings and comments. This density was a poor consideration on my part and has now been determined to be 350,000 plants-hm-2.

Results: I don't quite understand an issue related to the results of the field tests. Some of the results were discussed in years (soybean yield) and some only in 2022 what is the reason for this? This should be clarified and clearly stated in the manuscript. In addition, in this section there are statements that in my opinion should be in the discussion section

Answer: I rounded off some of the data from 2021 because of abnormal precipitation at the trial site in 2021. However, for soybean cultivation, yield is critical, so two years of yield is necessary. The discussion section has been revised and we have included a detailed discussion and explanation of the discussion section to highlight the interpretation of the study data for the final results. Once again, we thank the reviewers for their comments and guidance on this paper.

L337 Weather conditions were mentioned without any information about them. As I stated earlier I believe that they should be stated

Answer: We have provided relevant soil and two years of meteorological data for the area in the experimental design section to ensure that the experiment will provide readers with a wide range of references in subsequent future studies. Thank you to the reviewers for providing valuable comments, which have greatly improved my article.

Discussion

L380 In this subsection, except for one sentence, there is no description/explanation of the mechanisms that determined why the variation of fertilization between chemical and microbial influences the physical properties of the soil. When I read the manuscript, I was hoping for a broad description of the influence of the field experiment factors on the results obtained, but it is missing here. If the manuscript were to be published it is imperative that this be explained and supplemented. This is one of the key aspects for this manuscript.

Answer: A discussion section has been added to explain the individual data. We have included a detailed discussion and explanation of the discussion section to highlight the interpretation of the study data to the final results. I would like to thank the reviewers for providing valuable comments and I will continue to improve the paper writing.

L443 What are the findings?

Answer: References have been added and this section has been revised. Thanks to the reviewers for their comments and help.

L445 The reference to the literature must be added

Answer: Additional literature has been added, and I thank the reviewers for providing valuable comments, which have improved my article considerably.

L451 Why did this situation occur? What mechanisms determined this?

Answer: I have added the exploration of the mechanism of soybean yield components in the discussion section, thanks to the reviewer for providing valuable comments, we will continue to pay attention to it.

L462 Another aspect in favor of adding the weather results of the experiment. But there is also another question why microbial fertilizers affect plants due to different factors

Answer: We have provided relevant soil and two years of meteorological data of the area in the experimental design section to ensure that the experiment will provide readers with a wide range of references in subsequent future studies, have enriched the discussion section by detailing the specific mechanism of microbial fertiliser. Thanks to the reviewers' help, I feel that my professional knowledge has been greatly improved in the process of revising the paper.

L466 But why was this increased with a given fertilization?

Answer: Research on the effects of microbial fertilisers alone on crop growth and soil has been extensive and effective, however, the use of microbial fertilisers alone is economically expensive and the nutrient composition is difficult to maintain for subsequent crop growth and yield formation. In order to highlight the effects of microbial fertilisers in combination with chemical fertilisers, we have used a combination of the two, whereby more nutrients can be taken up by the plant, with a focus on the efficient use of nutrients and yield formation that can be achieved by the combination. We will continue our research in this direction in subsequent production experiments, and conduct comparative analyses with individual fertiliser applications. We thank the reviewers for their valuable comments, and we will continue to pay attention to them.

In addition, several Figures are very unreadable. The authors mentioned that they were prepared in MWord, I suggest improving them. Maybe add colors and increase their size?

Answer: Improved, thanks to the reviewer for providing valuable comments, which has improved my article a lot.

The References section is not uniform and not fully prepared properly, as are the references in the manuscript. I suggest you read the requirements for Authors and correct this.

Answer: I have adjusted the references and I thank the reviewers for their valuable comments, which have improved my article.

Round 2

Reviewer 2 Report

Authors have incorporate comments raised by reviewer. This manuscript may be accepted.

Author Response

Thanks again to the reviewer for all the help and improvement of this paper, and I wish you all the best.

Reviewer 3 Report

Dear Authors,

thank you very much for making changes to the manuscript after my suggestions. In my opinion, the manuscript has improved significantly. However, the description of the microbial fertilizer used in the manuscript (L 113 - 116) still lacks the composition of bacteria and fungi (despite the groups provided in the review response). In my opinion, a description similar to that in the review response should be given here, as well as the exact strains as well as their CFU. The research is based on the use of microorganisms so accurate reporting is extremely important.

Author Response

thank you very much for making changes to the manuscript after my suggestions. In my opinion, the manuscript has improved significantly. However, the description of the microbial fertilizer used in the manuscript (L 113 - 116) still lacks the composition of bacteria and fungi (despite the groups provided in the review response). In my opinion, a description similar to that in the review response should be given here, as well as the exact strains as well as their CFU. The research is based on the use of microorganisms so accurate reporting is extremely important.

Answer: Thank you very much for your comments and help, my dear reviewer. It was my negligence in not revising the manuscript, and I have now made another confirmation with the fertilizer company to confirm the composition of the compound microbial fertilizer as organic matter content ≥20% and total nutrient content (N+P2O5+K2O=8%, ratio 2:2:1). It is F01 (all-purpose) compound microbial fertilizer (containing Bacillus subtilis, Bacillus licheniformis, Bacillus spp., Bacillus megaterium, Bacillus coli, Bacillus spp., and carrier, etc., with the effective number of live bacteria ≥ 0.2 billion/g). This is all the information that the fertilizer company could provide me with, and I have written all the above information in the manuscript. Thanks again to the reviewer for all the help and improvement of this paper, and I wish you all the best.
